# Remote Sensing Image Scene Classification: Advances and Open Challenges

Ronald Tombe [1,*] and Serestina Viriri [2]

1 Computing Sciences Department, Kisii University, Kisii P.O. Box 408-40200, Kenya
2 School of Mathematics, Statistics and Computer Science, University of KwaZulu-Natal, Private Bag X54001, Durban 4000, South Africa
* Correspondence: ronaldtombe@kisiiuniversity.ac.ke

**Abstract:** Deep learning approaches are gaining popularity in image feature analysis and in attaining state-of-the-art performances in scene classification of remote sensing imagery. This article presents a comprehensive review of the developments of various computer vision methods in remote sensing. There is currently an increase of remote sensing datasets with diverse scene semantics; this renders computer vision methods challenging to characterize the scene images for accurate scene classification effectively. This paper presents technology breakthroughs in deep learning and discusses their artificial intelligence open-source software implementation framework capabilities. Further, this paper discusses the open gaps/opportunities that need to be addressed by remote sensing communities.

**Keywords:** deep learning; computer vision; artificial intelligence; contrastive learning; transformer-based learning; CNNs Feature extraction; geography-aware deep learning; datasets





## 1. Introduction

Remote sensing (RS) is an active research subject in the area of satellite image analysis for the discrete categorization of images into various scene category classes based on image content [1–3]. The satellite sensors periodically generate volumes of images that require effective feature processing for various computer vision applications, such as scene labeling [4], feature localization [5], image recognition [6], scene parsing [7], street scene segmentation [8], and many others. Several image feature analysis methods have been developed to this effect. References [2,3] groups the feature analysis methods into three categories: (a) low-level, which focuses on human-engineering skills, (b) medium-level, i.e., unsupervised methods that automatically learn features from images, and (c) high-level, i.e., deep learning methods that rely on supervised learning for feature analysis and representation.

The satellite-generated images vary in texture, shape, color, spectrum information, scale, etc. Additionally, remote sensing images exhibit the following characteristics:

- Complex spatial arrangements. Remotely sensed images have significant variations in the semantics (for instance, the scene images; of agriculture, airport, commercial areas, and residential areas are typical examples of varying scene image semantics). Extracting the semantic features from images requires effective computer vision techniques.
- Low inter-class variance. Some scene images are similar (e.g., agriculture and forest, dense residential areas, and residential-area). This characteristic is referred to as low intra-class variance. Achieving accurate scene classification under this circumstance requires well-calibrated computer vision techniques.
- High intra-class variance. Those same class scene images are commonly taken at varying angles, scales, and viewpoints. This diverse variation of same-class images requires well-designed computer vision approaches that can extract the same pattern features from the remotely sensed images regardless of their variations.

- Noise: Remotely sensed images are taken under varying atmospheric conditions and at different seasons. The scene images may have variable illumination conditions and require robust feature-learning techniques against varying weather circumstances.

Recent studies have indicated that data-driven deep learning models [2,9,10] attain state-of-the-art results in scene classification owing to their abilities in learning high-level abstract features from images. Developments in hardware for graphic processing units (GPUs) provide the capabilities to process the vast amount of data on deep learning frameworks. Deep learning [11] gives an architecture platform for feature learning methods that comprise several processing steps to learn remote sensing image features at different abstraction levels. Convolutional neural networks (CNNs) are good at abstracting local features and progressively expanding their receptive fields for more abstractions [12]. Transfer-based deep learning models [13,14] work on the premise that fundamental elements of images are the same; thus, they utilize pre-trained models that are trained on large-scale datasets for remote sensing applications. Other studies [15,16] develop models that fuse different CNNs in exploring their performances in scene classification. The application of transfer CNNs pre-trained models for feature extraction in remote sensing is limiting because they need to consider the features of remote sensing images. That is, remote sensing images are unique and vary in terms of background information, imaging angle, and spatial layout, factors that the CNNs pre-training models assume [17]. Successful CNN-based deep learning models in scene classification [9,18], object detection [19], and semantic segmentation [20–23] were incorporated into remote sensing subject to resolve the classic challenges efficiently since deep learning networks demonstrate to perform better in image classification, object detection, and semantic segmentation jobs.

Transformer-based deep learning methods such as the excellent teacher network guiding small networks (ET-GSNet) [24], and the label-free self-distillation contrastive with transformer architecture (LaST) [25] can learn long-range contextual information. An integration framework [26] combines vision-transformer and CNNs to attain impressive results with remote sensing public datasets. Although deep learning methods attain awe-inspiring results in scene classification and object detection, they must improve to deliver practical and scientific problems. First, deep learning methods rely on available datasets and do not utilize geography knowledge or features, often resulting in inaccurate predictions [27]. Second, lack of sufficient labeled datasets for training deep learning methods to generalize in new geographical regions [28]. Due to these challenges, new research directions in geography-aware deep learning models are emerging [27,28]. This research paradigm fuses knowledge and data in designing deeply blended deep learning models differently. Geography-aware deep learning is an emerging research area in remote sensing, and the research directions in this area include regional knowledge/features, physical knowledge/features, and spatial knowledge/features. The deep learning approaches for fusing geography knowledge and feature are majorly focusing on (1) rule-based, (2) semantic-networks, (3) object-based, (4) physical model-based, and (5) neural network-based [27].

The main contributions of this paper are as follows:

1. This survey presents image-feature analysis methods, strengths, and shortcomings.
2. This paper discusses the CNN architectures commonly adopted for the scene classification of remote sensing imagery.
3. We present the deep learning models that integrate both knowledge and data architectures, which are transformer-based and ontology-based models.
4. This paper discusses the advanced machine learning implementation frameworks; they are commonly utilized in implementing deep learning solutions in remote sensing image scene classification.
5. We discuss the properties of remote sensing datasets and their uniqueness in evaluating the different feature learning approaches.
6. This work presents the performance evaluation metrics upon which the feature analysis methods are evaluated to determine their scene classification effectiveness.

7.   This paper discusses the open opportunities that need to be addressed by the remote sensing community.

This paper is structured as follows: Section 2 discusses the various feature learning and analysis approaches from the context of remote sensing. Section 3 concisely discusses the deep learning architectures commonly adopted in remote sensing. Section 4 discusses, in brief, deep learning models that apply transformer-based learning techniques. Section 5 presents the popular loss functions: the softmax and hinge loss. Section 6 presents the popular open-source deep learning implementation frameworks for image classification and object detection, and Section 7 outlines the remote sensing datasets and their properties. Section 8 discusses the findings of this paper, while Section 9 concludes the article with insights into future works.

## 2. Image Feature Learning Approaches

Remote sensing image scene classification aims at annotating scene image patches to a semantic class depending on its contents. Figure 1 puts this concept into perspective. The feature-learning methods can be grouped into three categories: (a) pixel-based scene classification aimed at annotating every pixel to a category; (b) mid-level scene classification focused on identifying objects in remote sensing objects; (c) high-level scene classification, categorizing every remote sensing feature patch into a semantic category.

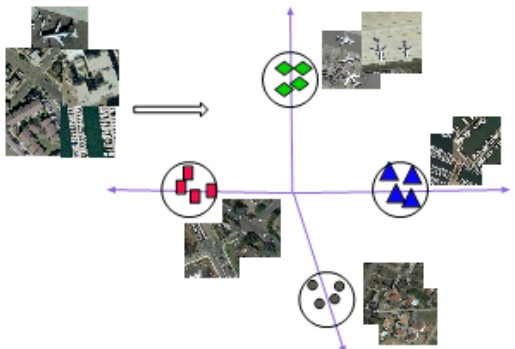

**Figure 1.** Scene classification of remote sensing images based on their features.

### 2.1. Pixel-Based Feature Learning Methods

2.1.1. Local Binary Patterns

This method identifies "uniform" local binary patterns (LBP) as critical attributes representing image texture. The uniform local binary patterns apply to generate occurrence histograms powerful for texture-feature representation [29]. LBPs characterize an image using the spatial information of the image texture structure. The LBP is calculated by thresholding neighbor $\{p_i\}_{i=0}^{n-1}$ pixels with the center $p_c$ pixel to compute an n-bit binary number, which is then converted to decimal as per Equation (1).

$$LBP_{n,r}(p_c) = \sum_{i=0}^{n-1} s(p_i - p_c)2^i = \sum_{i=0}^{n-1} s(d_i)2^i, s(x) = \{_{0,x<0}^{1,x>0} \qquad (1)$$

where $d_p = (p_c - p_i)$ denotes the difference among center and neighbor (P) pixels describing the spatial structure of the central location with a local difference vector $[d_0, d_1, \ldots, d_P - 1]$, LBP then generates a histogram as depicted in Equation (2).

$$(H(m)) = \sum_{p=0}^{P} \sum_{j=0}^{J} f(LBPs_{n,r}, m \in [0,m]), f(x,y) = \begin{cases} 1, \\ 0, \end{cases} \qquad (2)$$

Here $m$ denotes the maximum LBP pattern number.

LBP partitions an image into a fixed-size grid of cells to accomplish the pooling of local texture descriptors. The coarse quantization of spatial features is unrestricted, thus not well turned to the image morphology. This inevitably results in the loss of some discriminative information.

2.1.2. Multi-Scale Completed Local Binary Pattern

The multi-scale completed local binary pattern descriptor (MS-CLBP) [30] combines texture features in multiple scales sufficiently to cope with the limitations of a single-scale with the LBPs. MS-CLBP works similarly to LBP [29] where the circle radius, r, is modified to vary spatial image resolution. Combining operators CLBP_S (S is for sign) and CLBP_M (M is for magnitude) while varying(scale(m) and radius(r)) information parameter values achieves Multi-scale. Technically, this means combining CLBP_S and the CLBP_M histogram features extracted on each scale to form an MS-CLBP descriptor. The MS-CLBP method is applied in remote sensing [30], giving improved results on scene classification compared to LBP.

2.1.3. Distinctive Features Scale-Invariant

The distinctive feature scale-invariant (SIFT) [31] is a prominent descriptor in terms of distinctiveness. A single feature finds its correct match with great probability in a database of features. The SIFT constructs a feature-representation vector in four major steps that include:

1.  Scale-space extrema detection: a cascading algorithm identifies candidate points, which are further inspected. Once key candidate locations and scales were established that can be replicated with different views on the same object, the detection of locations follows that are invariant to scale changes of an image through looking for possible features across every probable scale. The Scale-space search algorithm accomplishes the task mentioned above.
2.  Localization of keypoint features: This is the process of establishing key candidate features by comparing neighbors to find a precise fit of the nearby data for location and scales of the primary curvatures. Points with low contrast (sensitive to noise) or unsatisfactorily localized along the edges are rejected.
3.  Orientation allocation: A consistent assignment is done for every keypoint depending on local image feature attributes in this step. The keypoint descriptor is formulated for orientation; therefore, they are invariant to image rotations. The keypoint descriptors then generate orientation histograms from gradient orientations based on sample points within a region surrounding the keypoint. A histogram contains 36 bins spanning 360 degrees of orientation. The gradient magnitudes determine every sample on the histogram, and a Gaussian-weighted circular window [31]. Dominant directions of local gradients are 'peaks' of orientation histograms. The highest peak detected in the histogram and other local peaks above 80% of the highest peak apply to generate a key point on an orientation histogram.
4.  Local image descriptor: The operations described above, i.e., (detection of scale-space, keypoint localization, and orientation assignment) assigned to an image the (scale, orientation, and location) for each keypoint. These parameters create a repeatable local 2D coordinate system that characterizes the local region of an image, thereby providing an invariant feature descriptor [31]

The experiment results [3] demonstrate that the SIFT method performs poorly when it is used as a low-level-visual feature descriptor on remotely sensed scene images.

*2.2. Mid-Level Feature Learning Methods*

2.2.1. The Bag of Visual Words

The bag of visual words (BoVWs) is an invariant statistical keypoints feature vector representation [32]. BoVWs quantize the patches generated with either the SIFT or by the LBP feature descriptor methods and then use the k-means algorithm to learn the holistic

scene image feature representations [3]. Equation (3)) shows the BoVWs workings to generate a histogram of the visual words, i.e., feature representation.

$$BoVWs = [t_1, t_2, \ldots, t_M],\tag{3}$$

where $t_m$ is the occurrence counts of features $m$ contained in an image and $M$ is the feature dictionary size. The BoVWs method is applied to scene classification [2,3] in several datasets, such as (the UC Merced, AID, and RESISC45) where it demonstrates better scene classification accuracy results compared to low-level methods such as the LBP and SIFT.

### 2.2.2. Fisher Vectors

A Fisher kernel coding framework [33] extends the BoVWs model to characterize the low-level features using a gradient rather than the count statistics in the BoVWs framework. This reduces the codebook size, accelerating the codebook's learning process. Experimental results by [33] on scene classification show that the Fisher kernel coding framework reduces the computational cost significantly to achieve better scene classification accuracy than methods based on the traditional BOVW model. Previous experiments [2,3] demonstrate this with different remote sensing datasets where the Fisher vector encodes low-level method features for image classification, and it achieves superior performance than traditional BoVWs.

### 2.3. High-Level Feature Learning Methods

### 2.3.1. Bag of Convolutional Features

The bag of convolutional feature (BoCF) [34] extracts image features in four stages, i.e., (1) convolution feature extraction, (2) codebook generation, (3) BoCF feature encoding, and (4) scene classification. The BoCFs use CNNs to automatically learn the image features, encoding histogram representations on every image. This is accomplished by quantizing every feature descriptor into visual words in the codebook. A linear classifier is a support vector machine (SVM) that learns the convolutional features fed to it for scene classification.

### 2.3.2. Adaptive Deep Pyramid Matching

The adaptive deep pyramid matching (ADPM) [35] method considers convolutional features as "multi-resolution deep-feature-representation" of the input image. The ADPM extracts image features of varying scales and then fuses them optimally. The ADPM extracts features from different layers in a deep spatial-pyramid matching manner following Equation (4). Assume $H_{l,1}$ and $H_{l,2}$ are the histograms of two images at layer $l$ (corresponding to different layer resolution), then, the feature match at layer $l$ is calculated as the histogram intersection with Equation (4)

$$I(H_{l,1}, H_{l,2}) = \sum_{l=1}^{L} w_l I(H_{l,1}, H_{l,2})\tag{4}$$

$L$ is the number of convolution layers, the fusing weight of the $l$th layer, and $I(H_{l,1}, H_{l,2})$ $= \sum_d^D min(\sum_{i,k} \delta(m_{1,l}^{(i,k)}, c_l^d), \delta(m_{2,l}^{(i,k)}, c_l^d))$.

### 2.3.3. Deep Salient Feature-Based Anti-Noise Transfer Network (DSFBA-NTN)

This technique comprises two significant steps, 1. deep salient feature (DSF) extraction step, which extracts scene patches utilizing visual-attention mechanisms. In this step, the salient regions and scales are detected from an image [36]. These features feed to a pre-trained CNN model to extract the DSF. In step 2, the anti-noise transfer network suppresses the effects of different scales and noises of scene images. The anti-noise network imposes a constraint to enforce training samples before and after inducing noise to scene images while learning the inputs of original scenes and different noises. The anti-noise network acts as a classifier.

2.3.4. Joint Learning Center Points and Deep Metrics

Conventional deep CNN with the softmax function can hardly distinguish the scene classes with great similarity [37]. To address this problem, the supervised joint-learning with the softmax hinge loss and the center-based organized learning metrics minimize the intra-class variances and maximize the inter-class variances of the remote sensing scene images, which results in better accuracy in scene classification.

## 3. Deep Learning Architectures

Deep learning architectures (which include: CNNs, autoencoders, and GANs) have demonstrated powerful capabilities to learn discriminant features and have penetrated several research areas, including the field of remote sensing image scene classification. Deep learning has attained the best scene classification accuracy. The deep learning architectures extract image features from low-level to high-level [1–3] and abstract these features for accurate scene classification tasks. Understanding the deep learning architecture properties is essential for developing remote sensing applications. This work surveys popular deep learning architectures that apply in remote sensing image scene classification. These include AlexNet, VGGNet, GoogLeNet, Inception, EfficientNet, U-net, deep residual networks, and DeepResUnet architectures. The subsequent subsections provide descriptions of the architectures.

### 3.1. Autoencoders and Stacked Autoencoders

Autoencoders (AEs) [38] comprise three layers: input, hidden, and output (Figure 2). It has two parts, i.e., the encoder and the decoder. The input layer transforms inputs into hidden layers, and the decoder performs the inverse, transforming the inner units into outputs. This is accomplished through non-linear mappings [1] where an input $x$ maps to a hidden, latent representation in a simple network through a function $h = f(Wx + \beta)$. $W$ is the weight matrix estimated during training, whereas $\beta$ is the bias vector. The mapping function is reconstructed (decoded) with $y = W^T h + \beta'$. The hidden units are less compared to the inputs or outputs; this is an essential feature of autoencoders. Thus, an autoencoder achieves dimensionality reduction via data comprehension through a hidden sparse autoencoder layer. Stacked autoencoders comprise multiple layers of AEs where each layer's output connects to the next layer's inputs. Provided with a training dataset $X = \{x_1, x_2, \ldots, x_N\}$, training a sparse autoencoder [39] works to find optimal parameters by minimizing the loss function in equation Equation (7). Autoencoders apply in the feature processing hierarchy. Autoencoders are applied in feature characterization [40–42] and have attained good results in remote sensing image scene classification.

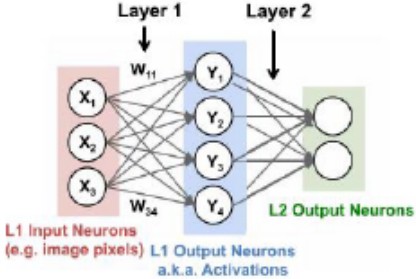

**Figure 2.** The architecture of autoencoder.

### 3.2. AlexNet

AlexNet [19] comprises five convolutional layers and three fully connected layers. Further, it contains normalization layers that follow the first and second convolution layers. Its Pooling-layers are placed immediately after the normalization layers and after the fourth convolutional layer. The AlexNet architecture won the ImageNet large-scale visual recognition challenge (ILSVRC) [43] in 2012. The AlexNet uses rectified linear units (ReLUs), which decreases the training time since ReLUs are faster than the hyperbolic

tangent function. It also implements the dropout layers to minimize the network overfitting problem. The AlexNet model is trained on the GPU; thus, it offers more cores than CPUs, allowing it to train larger image datasets faster. AlexNet applies in remote sensing, and it can attain magnificent results [2] in scene classification of remote sensing imagery.

### 3.3. VGGNet

The VGGNet [12] has two common architectures, i.e., VGG-19 and VGG-16. These two architectures are distinguishable by network depth, whereas the design is the same. The VGGNet strictly uses $3 \times 3$ filters with a padding and stride of 1, a max-pooling layer of $2 \times 2$ with a stride of 2. The VGGNet has the following properties:

- Its filters use receptive fields of size 3. These are smaller than AlexNet ($5 \times 5$ or $7 \times 7$).
- On the same blocks, they contain the exact size of feature maps and the number of filters in every convolutional layer.
- The size of features increases in deeper layers; they double after every max-pool layer.

The VGGNets are hierarchical feature characterization architectures of visual data, improving the classification accuracy [2,3,18] of remote sensing images. VGG-16 has thirteen convolutional layers, five-pool layers, and three fully connected layers. VGG-16 is commonly utilized for transfer learning in feature extraction of remote sensing imagery, for instance, in the works of [2,9,10], where it achieves impressive scene classification results.

### 3.4. GoogleNet

The main attribute of GoogLeNet [44] architecture is the improved efficiency in using computing resources within the network. The depth and width of the network are increased while retaining the computational-budget constant. GoogleNet architecture's main advantages include (1) utilization of different filter sizes in the same layer, which retains most of the spatial information; (2) parameter reduction with the network. Hence it is less susceptible to over-fitting and allowing it to be deeper. Comparing GoogleNet with AlexNet network, GoogLeNet has twelve times fewer parameters. GoogleNet achieved state-of-the-art object detection and classification tasks in the ImageNet Large-Scale Visual Recognition Challenge 2014.

Inception

The key concept of the inception network architecture [45] is that it incorporates sparsity by replacing fully connected layers with sparse layers within the convolutions. The "Inception-modules" are stacked layer-wise. Their output-correlation statistics vary because of high feature abstraction, captured with higher layers, hence decreasing spatial feature concentration. This requires feature embedding in the compressed, dense form to minimize the great dimensionality problem [44]. Reference [44] uses $1 \times 1$ convolutions for dimensionality minimization before applying the expensive $3 \times 3$ and $5 \times 5$ convolutions. Further, the inception architecture utilizes the rectified linear unit (ReLU) activation (Figure 3c) to enhance the network sparse feature representation. Figure 3 shows the different versions of the inception architecture applicable to computer vision tasks.

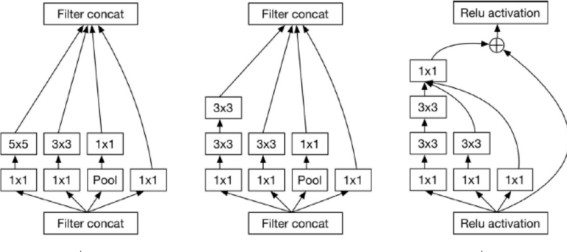

**Figure 3.** (**a**) Inception architecture block-V2 for dimension reduction [44], (**b**) inception architecture block-V3 designed for computer vision [46], (**c**) inception architecture block-V4 with short-cut connections [45].

### 3.5. EfficientNet

The EfficientNet [47] uniformly scales CNN's depth, width, and resolution with a compound co-efficient θ Figure 4. Instinctively, the parameter θ is specified by a user, and it controls the number of resources available to scale the network model. EfficientNet B3-aux [18] achieves state-of-the-art results in remote sensing with a low computational cost.

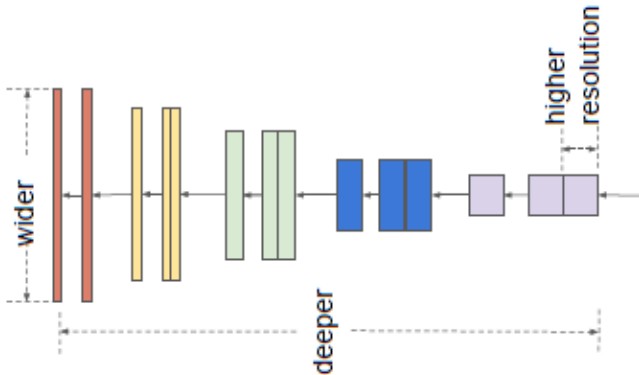

**Figure 4.** Compound scaling of CNN with EfficientNet [47].

### 3.6. Fully Convolutional Networks

The fully convolutional network (FCN) [48] performs semantic labeling to each pixel of an image, Figure 5. To perform semantic segmentation, the output of FCN has to be of the same pixel dimension as the input. Other properties of FCN include:

- Introduces skip connections to fuse information from different network depths to achieve multi-scale inferencing.
- Uses fully convolutional architecture model. This permits it to take arbitrary size images as inputs because in the absence of fully connected layers; no specific activation sizes are required at the end of the network.
- The FCN allows end-to-end learning through the encoder and decoder framework, which compresses and expands.

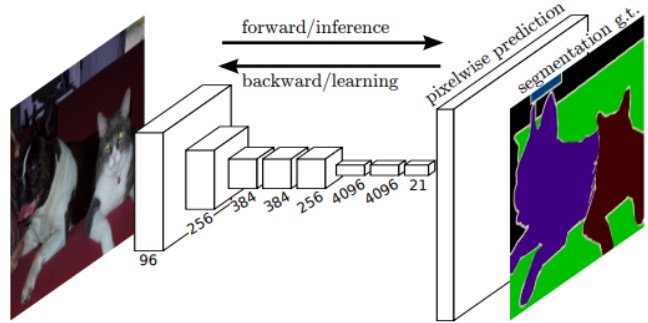

**Figure 5.** The Full Convolutional Framework.

### 3.7. U-Net

The U-Net network [20] performs data augmentation to utilize annotated image samples efficiently. It comprises a contraction path and an expansion path. The contraction section follows the conventional architecture of a convolutional neural network; that is, it consists of two 3 × 3 convolutions stacked with each other, followed by a nonlinear rectified unit and a 2 × 2 pooling operation of stride 2 for down-sampling. The number of feature channels is doubled at every down-sampling step. The expansion section comprises up-sampling steps for feature maps in every step followed by a 2 × 2 up-convolution that reduces by half the number of feature channels, then a concatenation of cropped feature

maps from the corresponding contraction section and two $3 \times 3$ convolutions which are followed by a nonlinear rectified unit. On the last layer, a $1 \times 1$ convolution applies in mapping every 64-feature component vector to required classes. The network has a total of 23 convolution layers. A U-Net variant [21] provides high scalability with short processing time while it achieves good accuracy results in remote sensing image scene classification.

### 3.8. Deep Residual

The increase in depth of deep networks results in network degradation because of the vanishing gradient problems. In addressing this issue, the deep residual architecture [6] is designed to ensure that the gradient propagates directly from the top to the bottom of the deep network over the backpropagation. Consider a mapping function $H(x)$ that fits some stacked layers. In this case, $x$ depicts the input of the first layers. Several nonlinear layers can estimate functions asymptotically. Equally, they can asymptotically approximate residual functions, for instance, $H(x) - x$ (for the same input and output dimensions) rather than stacked functions approximating $H(x)$, they estimate a residual function $F(X) := H(x) - x$. Reformulating this to $F(x) + x$ (a form that is equivalent to the original function $H(x)$). This reformulation $F(x) + x$ resolves the network degradation problem. This implies that when added network layers are configured as identity mappings, more deep models should contain the same training errors as the shallower counterpart network models. The residual learning algorithm is formulated as follows:

$$y = F(x, \{W_i\}) + x. \tag{5}$$

The vectors $x$ and $y$ are for the input and output of the layers, which are considered, and $F(x\{W_i\})$ is the residue mapping function that is learned. With stacked layers where they are two or more, $F = W_2\sigma(W_1x)$, here $\sigma$ represents a ReLU. A residual mapping operation $F + x$ is performed through short-cut connections with element-wise addition. Following stacked layers with short-cut connections when the dimensions of $F$ and $x$ are different, a linear projection $W_s$ is conducted with the short-cut connections to achieve dimension matching:

$$y = F(x, \{W_i\}) + W_sx. \tag{6}$$

The function $F(x, \{W_i\})$ depicts several convolutional layers.

Literature studies [49] indicate that residual architecture significantly enhances the scene classification accuracy of remote sensing imagery with increased network depth.

### 3.9. DeepResUnet

The DeepResUnet [23] integrates U-Net and deep residual networks. It has two sub-networks: A down-sampling network for feature maps extraction and an up-sampling network to reconstruct the extracted feature maps back so the input image's original size. This network results in low network degradation in the trained model since it utilizes a deep residual learning strategy. The segmentation results at the final layer are classified with the softmax classifier.

### 3.10. Unified Multimodal Data Analysis Deep Learning Architecture

The multimodal deep learning remote sensing (MDL-RS) [50] architecture integrates joint modalities: pixel-wise and spatial–spectral of fully connected networks and convolutional neural networks to characterize the scene with more detail and precision than using single modality data. Each of these modules focuses on feature characterization learning of multimodal data with the *extraction network (Ex-Net)* and *fusion network (Fu-Net)*. Figure 6 shows MDL-RS architecture. The MDA-RS jointly trains the two sub-networks (extraction network and fusion network) from end to end.

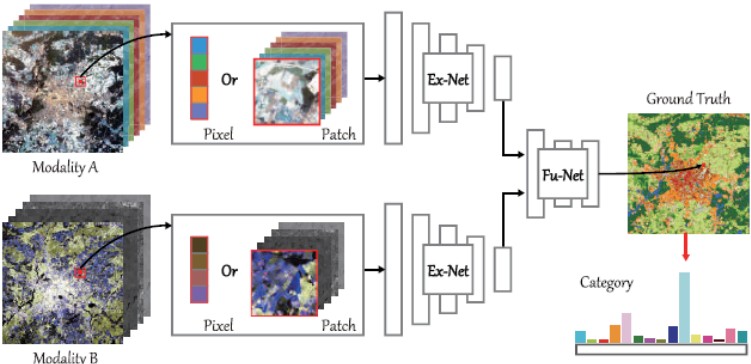

**Figure 6.** The multimodal deep learning remote sensing architecture for scene classification with sub-networks: *Extr-Net* and *Fu-Net*. The *Extr-Net* comprises two feature extractors, namely: pixel-wise fully connected network and the spatial–spectral convolutional neural network [50].

## 4. Transformer-Based Deep Learning Models

### 4.1. Excellent Teacher Network Guiding Small Networks (ET-GSNet)

The ET-GSNet [24] comprises two components, the teacher (vision transformer) and student models. The teacher model learns the feature relationships within the high spectral resolution image patches in the ET-GSNet. The student model applies ResNet18 in learning local features. These networks work in two optimized phases, with the teacher model transmitting long-range dependency knowledge from the vision transformer to ResNet18, Figure 7 depicts this process.

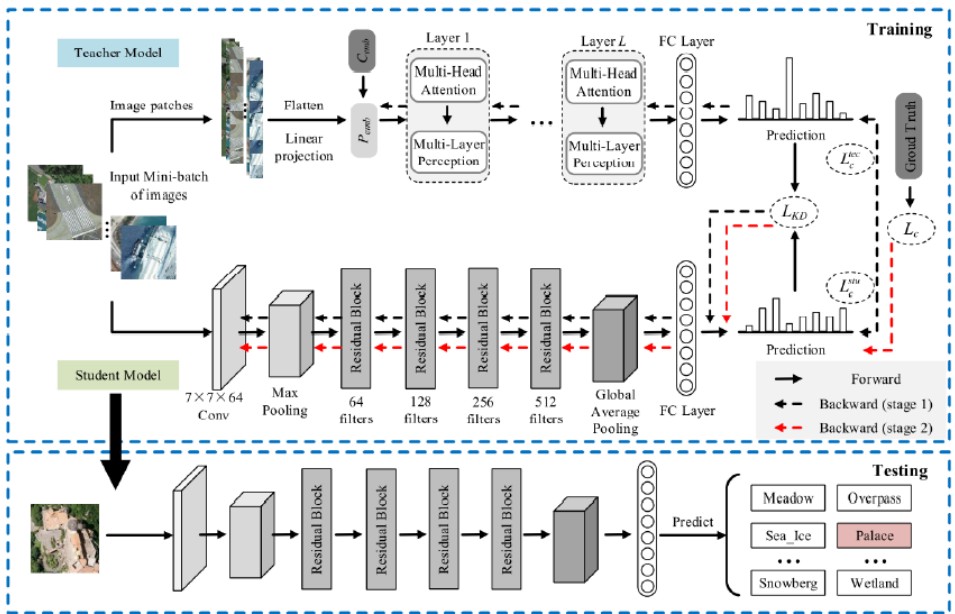

**Figure 7.** Excellent teacher network guiding small networks (ET-GSNet) [24].

### 4.2. Label-Free Self-Distillation Contrastive Learning with Transformer Architecture (LaST)

The Label-free self-distillation contrastive learning with transformer architecture (LaST) [25] uses knowledge distillation to gain global (long-range) knowledge of scene images which it fuses with the local features of the target student network. Figure 8 shows this process. LaST comprises the backbone and head, i.e., the backbone does feature extraction representations for the downstream tasks while predictions by the softmax project to the head.

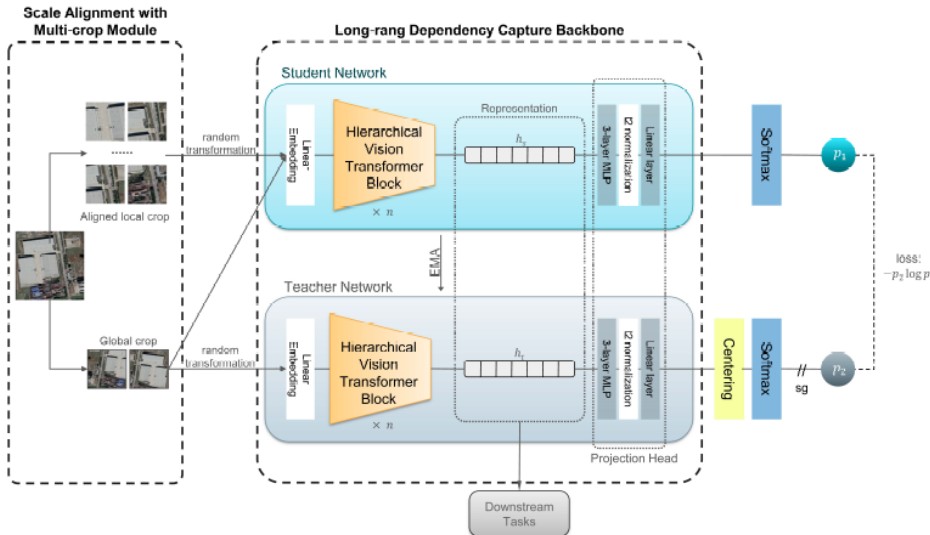

**Figure 8.** Label-free self-distillation contrastive learning with transformer architecture(LaST) [25].

## 5. Machine Learning Algorithms

### 5.1. The Softmax Function

Softmax [51] is a popularly utilized loss function with CNNs. Given a train set $[\mathbf{x}^{(h)}, \mathbf{y}^{(h)}; h \in 1, \ldots, N, 1 \in \mathbf{y}^{(h)} 1, dots, C]$, notice that $\mathbf{x}^{(h)}$ represents the $h$th patch of the image input, whereas $\mathbf{y}^{(h)}$ is its target class label from the $C$ classes.

A prediction for the $i$th class based on the $h$th input is a transformation by the softmax loss function: $p_i^{(h)} = e^{z_i^{(h)}} / \sum_{l=1}^{C} e_l^{z(h)}$, in this case, $z_i^{(h)}$ is typically the activation functions of a densely connected network layer. Therefore $z_i^{(h)}$ can be expressed as $z_i^{(h)} = \mathbf{w}_i^T \mathbf{f}^{(h)} + \mathbf{b}_i$. The softmax transforms predictions to give the probability distributions for the $C$ classes. These probabilistic kinds of predictions apply to computing the multinominal logistic loss. The softmax loss is defined as:

$$L_{softmaxloss} = -\frac{1}{N} \{ \sum_{h=1}^{N} \sum_{i=1}^{C} \mathbf{1}[\mathbf{y}^{(h)} = i] log p_i^{(h)} \} \tag{7}$$

The large-margin softmax loss function [52] advances the softmax loss by incorporating the angle margin for the angle $\theta_i$ in between an input vector $\mathbf{f}^{(h)}$ and the $i$th column $\mathbf{w}_i$, which is a weight matrix. Their investigation [52] shows that large-margin softmax loss is effective in avoiding overfitting, and it achieves better results on CIFAR-10, CIFAR-100, and MNIST datasets than the soft-loss function.

### 5.2. The Hinge-Loss Function

The hinge-loss function typically applies in training classifiers with large margins, e.g., the support vector machines (SVMs). The hinge-loss function for multi-class SVM is mathematically defined as:

$$L_{hinge-loss} = \frac{1}{N} \sum_{s=1}^{N} \sum_{r=1}^{C} [max(0, 1 - \delta(\mathbf{y}^{(s)}, r) \mathbf{w}^T \mathbf{x}_s)]^p \tag{8}$$

In this case $\delta(\mathbf{y}^{(s)}, r) = 1$ when $\mathbf{y}^{(s)} = $ r, else $\delta(\mathbf{y}^{(s)}, r) = -1$. Observe that when $p = 1$, Equation (8) is the hinge loss function ($L_1 - loss$), whereas if $p = 2$, it is a squared hinge loss function ($L_2 - loss$) [53]. The investigation [54] compares the performance of $L_2$ of SVMs with softmax in deep networks. The performance results on the MNIST dataset [55] show that $L_2 SVM$ is superior to the softmax function.

## 6. Deep Learning Open-Source Frameworks

Several open-source frameworks [56,57] provide artificial intelligence and advanced machine learning capabilities to implement deep learning. The following are popular frameworks in the remote sensing literature:

### 6.1. TensorFlow

TensorFlow [58] is an advanced machine learning system that works in heterogeneous environments and at large scales. TensorFlow provides dataflow graphs that apply to computation operations and shared states. The nodes of the dataflow map to different machines across clusters, including GPUs, multi-core CPUs, and tensor processing units. This framework gives application developers the flexibility to experiment with training algorithms. TensorFlow was released under an open-source license in 2015. The API of TensorFlow includes Python and C++. TensorFlow supports image, speech, handwriting recognition, natural language processing, and forecasting. TensorFlow is a popular deep learning implementation framework for remote sensing image scene classification [9,50].

### 6.2. Caffe

Caffe [59] is a popular deep-learning framework for the computer vision community. In 2014, It won an ImageNet Challenge. The Caffe framework offers deep learning tool kits for model training and deployment. Reference [35] attains state-of-the-art remote sensing scene classification results with the Caffe implementation framework. Caffe is C++ based and can be compiled on heterogeneous devices. Caffe supports Matlab, C++, and Python programming interfaces. The Caffe framework's vast user community contributes to the deep net repository called the "Model Zoo". GoogleNet and AlexNet are two standard user-made networks available to the public.

### 6.3. Deeplearning4J

The Deeplearning4J [60] framework has built-in GPU support, an essential feature for the training process, and supports Hadoop's distributed YARN, application framework. Deeplearning4J has a rich set of deep network architecture support: Recurrent Neural Networks (RNN), RBM, long short-term memory (LTSM) network, DBN, convolutional neural networks (CNNs), and RNTN. Deeplearning4J further provides support for a vectorization library known as Canova. Deeplearning4J is Java implemented and is faster than Python. This framework offers natural language processing, image recognition, and fraud detection capabilities.

## 7. Remote Sensing Datasets for Models-Evaluation

This section reviews the properties of popular remote sensing image datasets. Table 1 Gives the summary for different image datasets: (scene image classes, scales, cumulative images per class, the year of release, etc). Figure 9 shows the semantics of sample images taken from the recent remote sensing RESISC45 dataset [2].

**Table 1.** Summary of the remote sensing dataset properties.

| Datasets | ImagesPerClass | Classes | TotalImages | Resolutions(m) | Dimensions | Release YR |
|---|---|---|---|---|---|---|
| UC Merced [32] | 100 | 21 | 2100 | 0.3 | $256 \times 256$ | 2010 |
| WHURS [61] | $\approx 49$ | 19 | 1005 | $\approx 0.5$ | $600 \times 600$ | 2012 |
| RSSCN7 [62] | 400 | 7 | 2800 | – | $400 \times 400$ | 2015 |
| Aerial Image Dataset (AID) [3] | 220–420 | 30 | 10,000 | 8 to 0.2 | $600 \times 600$ | 2017 |
| RESISC45 [2] | 700 | 45 | 31,500 | 30 to 0.2 | $256 \times 256$ | 2017 |

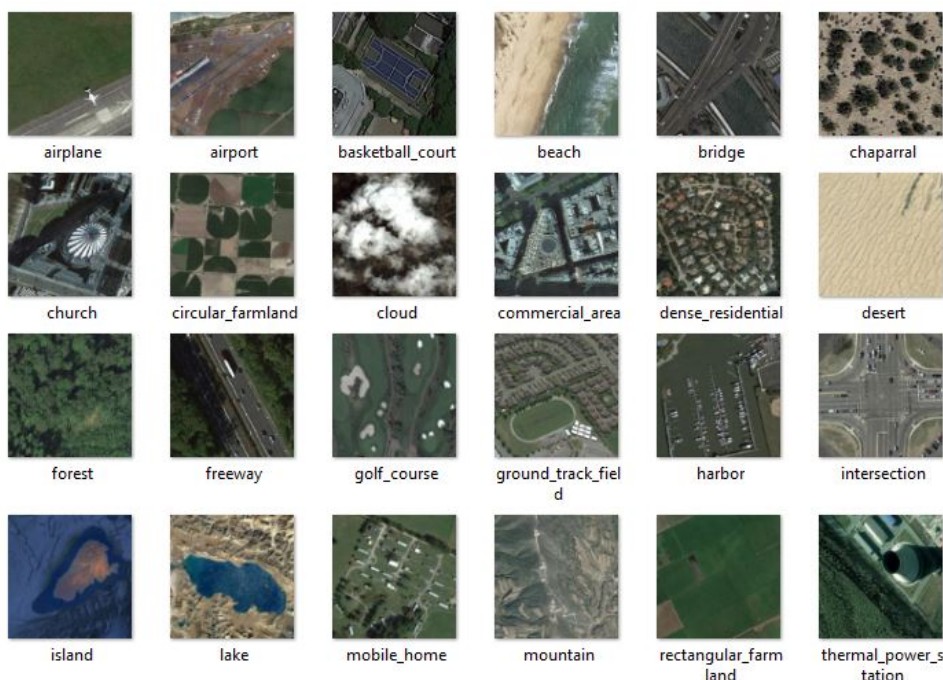

**Figure 9.** Semantics of sample images from RESISC45 dataset.

### 7.1. UC Merced Dataset

UC Merced is a 21-class dataset [32], each class contains 100 images of $256 \times 256$ pixel regions with a pixel resolution of 1 ft. This dataset comprises 2100 images that are uniformly labeled to 21 classes that are highly overlapped; for instance, (agricultural and forest are different by the type of vegetation covers). These diverse scene patterns present a challenge for effective feature characterization methods.

### 7.2. WHURS Dataset

The image size of the WHURS dataset [61] is $600 \times 600$ pixels with resolution up to half a meter. There are three release versions of this dataset. The initial version [61] contains 50 samples in each of the 12 aerial land cover classes. In a later version, ref. [63] extended the dataset to 19 classes. The new inclusions to this dataset are the [*desert, beach, football field, farmland, park, railway station, and mountain*] resulting in a total of 950 images. Reference [64] expanded WHURS to contain 5000 aerial images, and each class contains more than 200 image samples. WHURS dataset images are in different orientations and scales under various lighting conditions. The images are selected from numerous places around the globe.

### 7.3. RSSCN7 Dataset

The RSSCN7 dataset [62] has 2800 scene images obtained from Google Earth. It has 7 scene categories with each class containing 400 images of dimensions $400 \times 400$ pixels. This dataset's images have 4 different scales each, i.e., 100 images per scale in varying imaging angles. This renders the RSSCN7 dataset challenging properties that require well-crafted image analysis and feature representation methods that can work across the multiple-image scales and angles for robust feature analysis and interpretation.

### 7.4. Aerial Image Dataset

The aerial image dataset [3] is a more recent and relatively large dataset with 30 scene classes. Each class contains images ranging from 220–440 and a total of 10,000 images. The images are obtained from multiple sources with Google Earth imaging sensors across the world. AID images are multi-resolution, i.e., pixel resolutions range from 8 m to half-meters

at a size of 600 × 600. Key properties of AID, (1) high intraclass variations: due to the high spatial resolutions; therefore, geometrical structures of scenes are clearer; (2) smaller inter-class dissimilarity: AID has scenes that share similar objects, for example, both the stadium and playground can contain fields; (3) AID has a relatively large-scale dataset compared to RSSCN7, UC Merced, and WHU-RS. This dataset requires superior and more sophisticated image feature–descriptor methods for effective feature characterization.

### 7.5. RESISC45 Dataset

The RESISC45 [2] dataset has 31,500 images categorized into 45 classes of remote sensing imagery. Every class contains 700 images of dimensions 256 × 256 pixels in 3 color channels with spatial resolutions of about 30 to 0.2 m in every pixel. The RESISC45 dataset covers more than 100 countries and regions globally. The RESISC45 properties dataset (1) *is a huge dataset of remotely sensed images*; (2) *contains images with very diverse semantics*, i.e., the images chosen from different seasons under varying weather and illuminations conditions on different scales and resolutions; (3) *has huge inter-class similarity and intra-class diversity*. Therefore, the RESISC45 dataset is challenging, necessitating innovative (and well-calibrated) computer vision image analyses and interpretation techniques for effective image feature learning for the accurate scene classification of remote sensing imagery.

### 7.6. Metrics Performance Evaluation

#### 7.6.1. Overall Accuracy

The overall classification accuracy (OA) is the number of accurately classified sample images irrespective of the class they belong to, divided by the number of sample images

$$OA = \frac{accuratelyClassifiedImages}{SampledImages} \times 100 \tag{9}$$

#### 7.6.2. Average Accuracy

The average accuracy (AA) is the classification accuracy of every class irrespective of the number of image samples.

$$AA = \frac{accuratelyClassifiedImages}{TotalImages} \times 100 \tag{10}$$

#### 7.6.3. Confusion Matrix

The confusion matrix is a table that presents and analyses the errors and any resultant confusions among the various classes that are generated via counts of every type of incorrect and correct scene classifications by the test samples and aggregating the results on a table.

### 7.7. Scene Classification Performance Analysis of the State-of-the-Art

Table 2 presents the overall and average performance accuracies of the feature analysis methods, i.e., LBP, SIFT, MS-CLBP, BoVWs + SIFT, FV + SIFT, ADPM, DSFBA-NTN, JL-CPDM, and BoCFs with the different remote sensing datasets. Table 2 shows that high-level (deep learning-based) methods attain the best results with all four remote sensing datasets.

Table 3 shows the performance accuracies with different CNNs under training ratios of 80%, 50%, 20%, and 10%. It can be observed that the training-used ratio considerably affects the accuracy. VGG16, with a training ratio of 80%, achieves better accuracy than when the training ratio of 50% is used with the UC Merced dataset. This trend is the same as the AID and RESISC45 datasets. Table 3 also shows the CNN parameter settings.

**Table 2.** Scene classification with feature analysis methods.

| UC Merced Dataset | | |
|---|---|---|
| feature analysis Method | Method Level | accuracy % |
| LBP [2,3,65] | low-level | 36.29 ± 1.90 |
| SIFT [3] | low-level | 32.10 ± 1.95 |
| MS-CLBP [30] | low-level | 89.9 ± 2.1 |
| BoVWs(SIFT) [3] | Medium-level | 74.12 ± 3.30 |
| FV(SIFT) [3] | Medium-level | 82.07 ± 1.50 |
| ADPM [35] | High-level | 94.86 |
| DSFBA-NTN [36] | High-level | 98.20 |
| JLCPDM [37] | High-level | 97.30 ± 0.58 |
| **WHURS19 Dataset** | | |
| feature analysis method | Method Level | accuracy % |
| SIFT [3] | low-level | 27.21 ± 1.77 |
| BoVWs(SIFT) [3] | medium-level | 80.13 ± 2.01 |
| FV(SIFT) [3] | medium-level | 86.95 ±1.31 |
| ADPM [35] | High-level | 84.67 |
| DSFBA-NTN [36] | High-level | 97.90 |
| **AID Dataset** | | |
| feature analysis method | Method Level | accuracy % |
| LBP [3] | low-level | 29.99 ± 0.49 |
| SIFT [3] | low-level | 16.76 ± 0.65 |
| BoVWs(SIFT) [3] | Medium-level | 67.65 ± 0.49 |
| FV(SIFT) [3] | Medium-level | 77.33 ± 0.37 |
| **Resisc45 Dataset** | | |
| Feature analysis method | Method Level | accuracy % |
| LBP [2] | low-level | 21.74 ± 0.14 |
| BoVWs(SIFT) [2] | medium-level | 44.13 ± 2.01 |
| BoCFs [34] | High-level | 84.32 |

**Table 3.** Scene classification of remote sensing images by the fine-tuned CNNs approaches in the literature.

| UC Merced Dataset | | | | |
|---|---|---|---|---|
| Literature work | parameter settings | CNNs | accuracy % | Train % |
| [2] | Adam, learning rate (lr) = 0.001 Iterations= 1000–15,000 strides = 1000 | AlexNet + SVM | 94.58 | 80 |
| | | GoogleNet + SVM | 97.14 | |
| [66] | SGD, lr = 0.0001, iter = 50 | VGG16 | 97.14 | 80 |
| | | VGG16 | 96.57 | 50 |
| [18] | RMSprop, lr = 0.0001 | EfficientNet-B3 | 98. 22 | 50 |
| [18] | RMSprop, lr = 0.0001 | inception-v3 | 95.33 | 50 |
| **AID Dataset** | | | | |
| Literature work | parameter settings | CNNs | accuracy % | Train % |
| [18] | RMSprop, lr = 0.0001 | inception-v3 | 90.17 | 20 |
| [18] | RMSprop, lr = 0.0001 | EfficientNet-B3 | 94.19 | 20 |
| [66] | SGD, lr = 0.0001, iter = 50 | VGG16 | 93.60 | 50 |
| | | VGG16 | 89.49 | 20 |
| **Resisc45 Dataset** | | | | |
| Literature work | parameter settings | CNNs | accuracy % | Train % |
| [2] | | VGG16 | 84.56 | 10 |
| [10] | | VGGNet16 | 87.15 | 10 |
| [9] | | VGG16 | 91.05 | 15 |
| [10] | | VGGNet16 | 90.36 | 20 |
| [18] | RMSprop, lr = 0.0001 | EfficientNet-B3 | 91.08 | 10 |

## 8. Discussions

The tremendous advances in remote sensing sensor technologies over the past decade yield enormous image data for intelligent earth monitoring, such as scene classification of remote sensing imagery. Various computer vision techniques have been advanced in the literature (Table 2) to aid in feature analysis and subsequent scene interpretation. In particular, deep learning-based feature analysis has demonstrated state-of-the-art performance with the different remote sensing datasets. These scene classification approaches are generally dictated by the nature of the problem they address, such as the deep salient feature-based anti-noise transfer network (DSFBA-NTN) [36] attempts to address the noise problem. Even so, its primary operation mechanism works with patch-based feature extraction. Similarly, the other deep feature learning techniques—such as the adaptive deep pyramid matching [35], joint learning center points, and deep metric [37] methods—learn features in multi-layers, end up with high-feature representations that feed into a softmax [51] or an SVM [53], and solve pattern recognition problems (scene classifications, in this case).

It is also evident from Tables 2 and 3 that the performances of deep learning approaches substantially decrease with sizeable remote sensing datasets (specifically for the AID and the RESISC45 datasets). The implication is that deep learning approaches need to be well-calibrated to attain higher scene classification accuracy results with the more challenging remote sensing datasets; then, the knowledge will be migrated to real-time remote sensing artificial intelligence applications.

With deep learning architecture, VGG16 [12] is the most adopted scene classification of remote sensing images. This architecture is deep and efficient in its computing resource utilization, while its internal structure parameters are better able to calibrate the architecture for more effective image feature learning. In our paper [9], we utilize the VGG16 architecture for deep co-occurrence feature learning, then apply the ensemble classifiers for scene classification. M. Tan and Q.V. Le [47] developed a compound scaling mechanism for CNN; the method scales three network parameters: depth, width, and resolution. Reference [18] utilizes compound scaling with CNNs to attain state-of-the-art results in the scene classification of remote sensing imagery with UC Merced, AID, and RESISC45 datasets at low computation costs.

Recently, ref. [50] proposed a unified multimodal deep learning remote sensing (MDL-RS) architecture that integrates common modalities: pixel-wise and spatial–spectral aspects of fully connected networks and convolutional neural networks to characterize the scene with more detail and precision than using single modality data. The MDA-RS provides mechanisms to address the following challenging research questions in the field of remote sensing: "how to fuse", "what to fuse", and "where to fuse".

## 9. Conclusions and Future Work

Scene classification of remote sensing images aspires to annotate them to a semantic class based on their contents. We present a concise and comprehensive survey of the literature's feature representation methods in this view. This survey paper establishes that deep learning approaches attain superior accuracy in the scene classification of remote sensing images. Further, this paper covers the deep learning methods that have been applied, their strengths, and their shortcoming. Additionally, it covers the various deep learning architectures and available software implementation frameworks. This paper also presents evaluations from the literature on the effectiveness of deep learning approaches in scene classification with different remote sensing datasets.

In general, this work gives insights into the feature learning methods, deep learning architectures, and software frameworks that can be exploited for implementing deep learning solutions for remote sensing image scene classification. Additionally, this paper gives descriptions of remote sensing datasets that can be utilizable to evaluate the effectiveness of image semantic categorization strategies.

**Author Contributions:** R.T. and S.V. together conceptualized and wrote this paper; R.T.—draft preparation; S.V.—review and and mentorship. All authors have read and agreed to the published version of the manuscript.

**Funding:** This research received no external funding.

**Institutional Review Board Statement:** Not applicable.

**Informed Consent Statement:** Not applicable.

**Data Availability Statement:** Not applicable.

**Conflicts of Interest:** The authors declare no conflict of interest.

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
