# Peer review of "Remote Sensing Image Scene Classification: Advances and Open Challenges"

_2673-7418, doi:10.3390/geomatics3010007_

Round 1

Reviewer 1 Report

Summary Comments:

The subject of the review of the developments of various computer vision methods in remote sensing presenting technology breakthroughs in deep learning approaches in scene classification and discussing their artificial intelligence open-source software implementation frameworks’ capabilities is very signficance. However, it is not comprehensive in the sense of not having explored scene classification of Sentinel which is the state-of-the-art of public satellite images.

The manuscript is well organized. The title reflects the content of the work. The Abstract is less than 250 words and it is suitable. The keywords are representative of the work, but some are in the title and should be changed. The introduction is restricted to the subject of the work. The objective is clear and appropriate. Various computer vision methods in remote sensing area was presented and the deep learning-based feature analysis has demonstrated state-of-the-art performance with the different remote sensing datasets. The tables and figures are necessary and the captions are explanatory, but the Table is cropped. The bibliographic references are necessary. The text is a good English writing.

Specific Comments:

Line 10: The keywords "Remote Sensing" and "Scene Classification" are in the title that has already been indexed. These keywords must be changed for other keywords representative of the work.

Line 374: Table 1 is cropped on the page. Adjust.

Line 390: Include a topic of scene classification with Sentinel images. WorldCover (https://esa-worldcover.org/en) provides the global land cover products for 2020 and 2021 at 10 m resolution, developed and validated based on Sentinel-1 and Sentinel-2 data.

Author Response

Thanks to the reviewers for their insightful comments. 

Below is a detailed description of how we, the reviewer, concerns were used to improve the quality of the manuscript.

Reviewer 1

Comment 1

The subject of the review of the developments of various computer vision methods in remote sensing presenting technology breakthroughs in deep learning approaches in scene classification and discussing their artificial intelligence open-source software implementation frameworks' capabilities is very signficance. However, it is not comprehensive in the sense of not having explored scene classification of Sentinel which is the state-of-the-art of public satellite images.

Response to reviewer comment 1

Thanks for the insightful comment. In our revised work, we provide a course discussion on geography-aware deep learning models on remote sensing in lines 72-79 and the aspect of sentinel scene classification captured in general. Two references ([26] and [27]) are included for readers who may wish to get a deeper understanding. Note that because we wish to maintain the thought process/structure of the paper, introducing the topic and giving references will be more appropriate

Comment 2

 The manuscript is well organized. The title reflects the content of the work. The Abstract is less than 250 words and it is suitable. The keywords are representative of the work, but some are in the title and should be changed. The introduction is restricted to the subject of the work. The objective is clear and appropriate. Various computer vision methods in remote sensing area was presented and the deep learning-based feature analysis has demonstrated state-of-the-art performance with the different remote sensing datasets. The tables and figures are necessary and the captions are explanatory, but the Table is cropped. The bibliographic references are necessary. The text is a good English writing.

 Response to reviewer comment 2

We very much appreciate this observation: table 1 and table 3 have been adjusted accordingly. 

Specific Comments:

Comment 3

 Line 10: The keywords "Remote Sensing" and "Scene Classification" are in the title that has already been indexed. These keywords must be changed for other keywords representative of the work.

Response to comment 3

 Thanks for this observation. The keywords have been replaced with different ones, which include Deep Learning; Computer Vision; Artificial Intelligence; contrastive learning, Transformer-based learning, CNNs Feature extraction; Geography-aware deep learning; Datasets

Comment 4

Line 374: Table 1 is cropped on the page. Adjust.

Response to comment 4

We very much appreciate this observation. Table 1 has been adjusted accordingly 

 Comment 5

 Line 390: Include a topic of scene classification with Sentinel images. WorldCover (https://esa-worldcover.org/en) provides the global land cover products for 2020 and 2021 at 10 m resolution, developed and validated based on Sentinel-1 and Sentinel-2 data.

 Response to comment 5

Thanks for the insightful comment. In our revised work, we provide a course discussion on geography-aware deep learning models on remote sensing in lines 72-79 and the aspect of sentinel scene classification captured in general. Two references ([26] and [27]) are included for readers who may wish to get a deeper understanding. Note that because we wish to maintain the thought process/structure of the paper, introducing the topic and giving references will be more appropriate 

Reviewer 2 Report

This paper presents a survey of computer vision models in remote sensing, especially scene classification tasks. The paper is well-written and easy to follow. However, there are two main concerns from my perspective.

1. There are insufficient surveys for computer vision models in other remote sensing tasks except for scene classification.  I understand that authors may focus on the scene classification task. However, based on their claim, I suggest that they can give a coarse outline for most remote sensing visual recognition tasks, such as semantic segmentation, change detection, and object detection, and then focus on scene classification. I provide some classic literature for their reference, i.e.,

10.1016/j.rse.2021.112636 10.1109/MGRS.2017.2762307 10.1109/CVPR.2019.01270 10.1109/TGRS.2021.3110314 10.1109/CVPR46437.2021.00420 10.1109/ICCV48922.2021.01002

2. More advanced deep scene classification models, such as Transformer-based methods, should be added. They should be discussed in terms of advantages and disadvantages compared to CNN-based methods for a full understanding.

Overall, this paper is promising after a major revision if the authors can solve these concerns.

Author Response

Reviewer 2. 

This paper presents a survey of computer vision models in remote sensing, especially scene classification tasks. The paper is well-written and easy to follow. However, there are two main concerns from my perspective.

Comment 1

  1. There are insufficient surveys for computer vision models in other remote sensing tasks except for scene classification. I understand that authors may focus on the scene classification task. However, based on their claim, I suggest that they can give a coarse outline for most remote sensing visual recognition tasks, such as semantic segmentation, change detection, and object detection, and then focus on scene classification. I provide some classic literature for their reference, i.e.,

10.1016/j.rse.2021.112636 10.1109/MGRS.2017.2762307 10.1109/CVPR.2019.01270 10.1109/TGRS.2021.3110314 10.1109/CVPR46437.2021.00420 10.1109/ICCV48922.2021.01002

Geography-Aware Self Supervised Learning. 10.1109/ICCV48922.2021.01002

Response to reviewer2 comment 1

Thanks for this insightful comment. We have enhanced the introduction section of the paper, and include more details that broadly give course explanations on remote sensing visual recognition, including visual recognition tasks, semantic segmentation, and object detection. Particularly in lines 46-62. Additionally, references are provided for readers who wish to dive deeper into any of the topics of interest. Furthermore, we utilize the references recommended to enrich our paper, specifically between lines 72-78.

Comment 2

  1. More advanced deep scene classification models, such as Transformer-based methods, should be added. They should be discussed in terms of advantages and disadvantages compared to CNN-based methods for a full understanding.

Response to comment 2

Thanks for this insight. In lines 48-57, we discuss the advantages of CNNs and their limitations. In sections 4, 4.1, and 4.2, we present and discuss recent works on transformer-based methods. The operations mechanisms (teacher-student networks) are meant to overcome the shortcomings of CNNs. Additionally, two recent references ([23] and [24]) are included for readers who may wish to dig deep into the subject of Transformer-based methods

Round 2

Reviewer 2 Report

The authors have solved my major concerns.

I tend to accept this paper.